# NUISANCES VIA NEGATIVA: ADJUSTING FOR SPURIOUS CORRELATIONS VIA DATA AUGMENTATION

## ABSTRACT

There exist features that are related to the label in the same way across different settings for that task; these are semantic features or *semantics*. Features with varying relationships to the label are *nuisances*. For example, in detecting cows from natural images, the shape of the head is a semantic and because images of cows often have grass backgrounds but not always, the background is a nuisance. Relationships between a nuisance and the label are unstable across settings and, consequently, models that exploit nuisance-label relationships face performance degradation when these relationships change. Direct knowledge of a nuisance helps build models that are robust to such changes, but knowledge of a nuisance requires extra annotations beyond the label and the covariates. In this paper, we develop an alternative way to produce robust models by data augmentation. These data augmentations corrupt semantic information to produce models that identify and adjust for where nuisances drive predictions. We study semantic corruptions in powering different robust-modeling methods for multiple out-of distribution (OOD) tasks like classifying waterbirds, natural language inference, and detecting Cardiomegaly in chest X-rays.

## 1 INTRODUCTION

Relationships between the label and the covariates can change across data collected at different places and times. For example, in classifying animals, data collected in natural habitats have cows appear on grasslands, while penguins appear on backgrounds of snow; these animal-background relationships do not hold outside natural habitats (Beery et al., 2018; Arjovsky et al., 2019). Some features, like an animal's shape, are predictive of the label across all settings for a task; these are semantic features, or semantics in short. Other features with varying relationships with the label, like the background, are nuisances. Even with semantics present, models trained via empirical risk minimization (ERM) can predict using nuisances and thus fail to generalize (Geirhos et al., 2020).

Models that rely only on the semantic features perform well even when the nuisance-label relationship changes, unlike models that rely on nuisances. Many methods exist to build models robust to changing nuisance-label relationships (Mahabadi et al., 2019; Makar et al., 2022; Liu et al., 2021; Puli et al., 2022; He et al., 2019); we call these spurious-correlation avoiding methods (SCAMs). These methods broadly fall into two classes: 1) methods that assume access to nuisances, like Nuisance-Randomized Distillation (NURD) (Puli et al., 2022), debiased focus loss (DFL), product of experts (POE) (Mahabadi et al., 2019), and 2) methods that rely on assumptions about ERM-trained models relying on nuisances, like Just Train Twice (JTT) (Liu et al., 2021). We point out a commonality between the two classes of methods: a model that predicts the label from the nuisance called a *biased model*, that are built using extra annotations or assumptions. Intuitively, biased models play a role in building robust predictive models by providing a way to detect when the nuisance can influence predictions.

How do we build biased models without extra annotations in the form of nuisances being known in the training data or assumptions about ERM-trained models relying on nuisances? In this work, we build robust models from a different and complementary source of assumptions: knowledge about semantics. Imagine using data augmentation to corrupt semantics in the covariates — if the resulting semantic-corrupted input can still predict the label, the prediction must rely on nuisances, thereby providing a window into nuisances that can be used to build a biased model.

Designing a data augmentation that corrupts semantics is easy. For example, replacing the covariates with random noise would fully corrupt the semantics. However, after such a corruption there

is nothing that predicts the label meaning no nuisance information would be identified. A better semantic corruption would corrupt the semantics, while preserving some nuisances. This preservation is possible when semantics and nuisances appear differently in the covariates; we call such a difference a separation. We identify different separations to develop semantic corruptions for object recognition and natural language inference (NLI).

The first separation is when semantics are global and nuisances are local. Formally, global semantics are position-dependent functions of the subsets of the covariates (patches in images, or words in sentences), while local nuisances are position-independent. For example, in recognizing cows, the shape of the animal structures the distant patches where the cow's eyes, ears, tail, hooves appear; nuisances like grass can appear anywhere without structure. Due to positional-dependence, randomizing positions of subsets of covariates corrupts global semantics; however, position-independent local nuisances are retained. Under the global/local separation, we corrupt semantics via patch randomization (PATCH-RAND) for images and n-gram randomization (NGRAM-RAND) for NLI.

The second separation is when certain parts of the input are required for semantics. For example in chest X-rays, lungs appear in the center, while nuisances like the scanner affect the border. In NLI, the premise sets up the context for detecting entailment. Without the premise, entailment cannot be determined by semantics, but the hypothesis retains some nuisances. For this separation, masking parts of the covariates corrupts semantics for object recognition via region-of-interest masking (ROI-MASK) and for the semantic context in NLI via premise masking (PREM-MASK).

The last two separations are when semantics and nuisances are signals of different frequencies or different pixel-intensities. For example, in detecting Cardiomegaly in chest X-rays, semantic features like the heart are low-frequency features with high pixel-intensity; see fig. 2. However, nuisances like noise due to the X-ray scanner can be high-frequency or low-intensity signals. For such separations, frequency filtering (FREQ-FILTER) and intensity filtering (INT-FILTER) corrupt semantics.

We demonstrate the value of semantic corruption by using it to power a variety of methods: NURD (Puli et al., 2022), DFL, POE (Mahabadi et al., 2019), and JTT (Liu et al., 2021). We run these methods by building biased models using nuisances produced by semantic corruption. These methods with semantic corruptions outperform ERM on out-of distribution (OOD) generalization tasks like waterbirds (Sagawa et al., 2019), cardiomegaly detection from chest X-rays, and NLI. The performance of NURD, DFL, POE run with semantic corruption is similar to what the methods achieve with extra observed nuisance variables. Finally, JTT with semantic corruptions outperforms vanilla JTT.

## 2 WHAT DO METHODS NEED TO REDUCE SPURIOUS CORRELATIONS?

A spurious correlation is a relationship between the covariates and the label that changes across settings like time and location (Geirhos et al., 2020). Models that exploit a spurious correlation can perform poorly outside the training distribution. We focus on the class of methods that correct models using knowledge of nuisances or where they might appear (Mahabadi et al., 2019; Liu et al., 2021; Puli et al., 2022); we call these spurious-correlation avoiding methods (SCAMs). With label $\mathbf{y}$, a vector of nuisances $\mathbf{z}$, and covariates $\mathbf{x}$, the goal is to predict well on data regardless of the nuisance-label relationship. Next, we establish that the central part of several SCAMs is a model that predicts the label using nuisances, which we call the biased model. Let $p_{tr}$ and $p_{te}$ be the training and test distributions respectively, and let $a \perp\!\!\!\perp b$ denote that the random variables $a, b$ are independent.

**NURD.** In tackling spurious correlations, Puli et al. (2022) identify a conditional that has performance guarantees on test distribution $p_{te}$ with an unknown nuisance-label relationship. They develop NURD to learn the conditional using data from $p_{tr} \neq p_{te}$. NURD uses 1) the nuisance-randomized distribution, $p_\perp(\mathbf{y}, \mathbf{z}, \mathbf{x}) = p(\mathbf{y})p_\perp(\mathbf{z})p(\mathbf{x} \mid \mathbf{y}, \mathbf{z})$, where $\mathbf{z} \perp\!\!\!\perp_{p_\perp} \mathbf{y}$, and 2) an uncorrelating representation $r(\mathbf{x})$ for which $\mathbf{z} \perp\!\!\!\perp_{p_\perp} \mathbf{y} \mid r(\mathbf{x})$. In $p_\perp$, the nuisance alone cannot predict the label; this helps avoid features that depend only on the nuisance. Next, features that are mixed functions of the label and the nuisance (e.g. $\mathbf{x}_1 = \mathbf{y} + \mathbf{z}$) can also be spurious. Uncorrelating $r(\mathbf{x})$ avoid such features. With these insights, NURD builds models of the form $p_\perp(\mathbf{y} \mid r(\mathbf{x}))$ that are most informative of the label. We work with reweighting-NURD, which estimates $p_\perp$ by weighting samples as $p(\mathbf{y})/p_{tr}(\mathbf{y} \mid \mathbf{z})p_{tr}(\mathbf{y}, \mathbf{z}, \mathbf{x})$. See appendix A for more details.

**End-to-end bias mitigation.** Mahabadi et al. (2019) consider two methods to train a biased model and a base predictive model jointly to make the base model predict without relying on the biases. The

**Table 1:** Summary of NURD, JTT, POE, and DFL. Each method approximates what we call a biased model: $p_{tr}(\mathbf{y} \mid \mathbf{z})$. This table describes the different biased models, their names, how they are built. We build biased models as $p_{tr}(\mathbf{y} \mid T(\mathbf{x}))$ where $T(\mathbf{x})$ is a semantic corruption.

| Method | Name | What the biased model is | Assumptions |
|---|---|---|---|
| JTT | Identification model | $p_{tr}(\mathbf{y} \mid \mathbf{x})$ learned via ERM | ERM builds biased models. |
| POE/DFL | Biased model | $p_{tr}(\mathbf{y} \mid \mathbf{z})$ learned via ERM | $\mathbf{z}$ from domain-knowledge. |
| NURD | Weight model | $p_{tr}(\mathbf{y} \mid \mathbf{z})$ learned via ERM | $\mathbf{z}$ from domain-knowledge. |

methods use and fine-tune a BERT model (Devlin et al., 2019) and do not propagate the gradients of the biased model to update the common parameters (token embeddings in this case). They propose 1) POE, where the `log` of the product of the predicted probabilities of the two models is used to compute the classification loss and 2) DFL, where the biased model is used to weight the cross-entropy loss for the base model. For both methods, Mahabadi et al. (2019) build a biased model as $p_{tr}(\mathbf{y} \mid \mathbf{z})$. The intuition is that the samples correctly classified by the biased model will have low loss and the base model focuses on classifying samples that the biased model misclassifies.

**Just Train Twice JTT.** With JTT, Liu et al. (2021) aim to build models robust to group shift, where the relative mass of a fixed set of disjoint groups of the data changes between training and test times. The groups here are subsets of the data defined by a pair of values of discrete label and nuisance values. While they work without relying on training group annotations, i.e. without nuisances, they assume ERM builds models with high worst-group error. JTT first builds an "identification" model via ERM to isolate samples that are misclassified due to reliance on the nuisances. Then, JTT trains a model via ERM on data with the loss for the misclassified samples upweighted (by constant $\lambda$). The number of epochs to train the identification model and the upweighting constant are hyperparameters that require tuning using group annotations (Liu et al., 2021).

**The commonality of a biased model.** The central part between NURD, DFL, POE, and JTT is a model that predicts the label using nuisances (like $p_{tr}(\mathbf{y} \mid \mathbf{z})$), which we call the biased model as in He et al. (2019); Williams et al. (2018). While these methods reduce dependence on nuisances, they build biased models using additional annotations or require assumptions that ERM-trained models predict using the nuisance. In the next section, we describe an alternative: corrupt semantic information with data augmentations to construct nuisances that can be used in a biased model.

## 3 ROBUSTNESS VIA SEMANTIC CORRUPTIONS

We define a data augmentation as a transformation of the covariates with random noise $\boldsymbol{\delta}$: $T(\mathbf{x}, \boldsymbol{\delta})$. Formally, an ideal semantic corruption is a data augmentation or transformation such that the label does not depend on the transformation given the nuisance $\mathbf{y} \perp\!\!\!\perp T(\mathbf{x}, \boldsymbol{\delta}) \mid \mathbf{z}$ and that the transformation is not independent of the nuisance $T(\mathbf{x}, \boldsymbol{\delta}) \not\perp\!\!\!\perp \mathbf{z}$. The first condition ensures that the biased model built from the semantic corruption only predicts the label because of the nuisance. The second condition ensures the semantic corruption depends on the nuisance. In designing a semantic corruption these two conditions are in tension. The former wants to destroy everything about the label unrelated to the nuisance, while the latter wants to retain everything about the nuisance, which may be hard to achieve without retaining extra information about the label. The design of a semantic corruption is made easier, when semantics and nuisances appear differently in the covariates; we call such a difference a separation. Focusing on two popular OOD tasks, object recognition and NLI, we identify separations and build semantic corruptions based on permutations and masking.

### 3.1 SEMANTIC CORRUPTIONS VIA PERMUTATIONS FOR A GLOBAL/LOCAL SEPARATION

The first separation we consider is between semantics that appear as global structure and nuisances that appear as local structure. We give an intuitive example for such global semantics and local nuisances before formalizing them. Consider the waterbirds dataset from (Sagawa et al., 2019) with waterbirds and landbirds appearing predominantly on backgrounds with water and land respectively. Semantic features like the wing shape and the presence of webbed feet are corrupted by randomly permuting small patches. However, the background nuisances remain after permutations because they can be detected from small patches due to colors and textures. See fig. 1a.

Formally, given subsets of the covariates $\mathbf{x}_1, \cdots \mathbf{x}_k$ extracted in an order, semantics $s(\mathbf{x}_1, \cdots, \mathbf{x}_k)$ change with the order of extraction while nuisances $n(\mathbf{x}_1, \cdots, \mathbf{x}_k)$ do not: for permutations $\Pi$

$$\exists \pi \in \Pi \quad s(\mathbf{x}_1, \cdots \mathbf{x}_k) \neq s(\mathbf{x}_{\pi(1)}, \cdots \mathbf{x}_{\pi(k)}), \qquad \forall \pi \in \Pi \quad n(\mathbf{x}_1, \cdots \mathbf{x}_k) = n(\mathbf{x}_{\pi(1)}, \cdots \mathbf{x}_{\pi(k)})$$

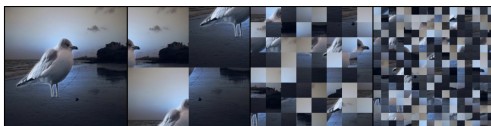 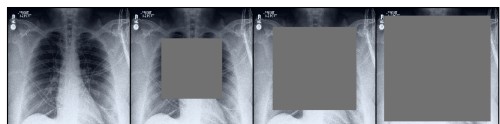

**(a)** PATCH-RAND to corrupt global semantics in Waterbirds. The original is the left most, followed by PATCH-RANDs with sizes $112, 28, 14$. At sizes less than $28$, shape is hard to make out.

**(b)** Masking to corrupt semantics in chest X-rays. The original is the left most, followed by ROI-MASK of size $112, 154, 196$. At sizes more than $154$, the heart is blocked out.

**Figure 1:** Semantic corruptions of Waterbirds via PATCH-RAND and chest X-rays via ROI-MASK.

We give a demonstrative example of a semantic corruption with global semantics and local nuisances. Consider a family of distributions $\mathcal{F} = \{p_D\}_{D \in R}$ with changing nuisance-label relationships. With $\mathcal{U}$ as the uniform distribution over $\{1, 2, 3\}$, and $\mathcal{N}$ as the normal distribution, samples from $p_D(\mathbf{y}, \mathbf{z}, \mathbf{x})$ come as $\mathbf{y} \sim \mathcal{U}$, $\mathbf{z} \sim \mathcal{N}(D\mathbf{y}, 1)$, and $\mathbf{y}$ selects a configuration of $\mathbf{x}$

$$\mathbf{y} = 1 \implies \mathbf{x} = [-\mathbf{z}, \mathbf{z}, \mathbf{z}], \qquad \mathbf{y} = 2 \implies \mathbf{x} = [\mathbf{z}, -\mathbf{z}, \mathbf{z}], \qquad \mathbf{y} = 3 \implies \mathbf{x} = [\mathbf{z}, \mathbf{z}, -\mathbf{z}]$$

The configuration is a semantic that determines $\mathbf{y}$ and computing it requires comparing coordinates: $\mathbf{y} = 1$ if $\mathbf{x}_2\mathbf{x}_3 > 0$, $\mathbf{y} = 2$ if $\mathbf{x}_1\mathbf{x}_3 > 0$, and $\mathbf{y} = 3$ otherwise. In words, the semantic feature is global. However, $\mathbf{z} = \mathbf{x}_1 + \mathbf{x}_2 + \mathbf{x}_3$, which means that $\mathbf{z}$ is determined regardless of where the negative sign is, i.e. it is local. Here, a random permutation $T(\mathbf{x}, \boldsymbol{\delta})$ of the coordinates in $\mathbf{x}$ is a semantic corruption: as $T(\mathbf{x}, \boldsymbol{\delta})$ permutes the location of the negation, $T(\mathbf{x}) \mid \mathbf{y}, \mathbf{z}$ is distributed identically to $T(\mathbf{x}) \mid \mathbf{z}$. Next, we give semantic corruptions for vision and language tasks with global/local separations.

**Patch randomization.** In image recognition, object recognition tasks where the object is a shape in the foreground and the color and texture in the background are nuisances, often satisfy the global-semantics and local-nuisances separation. For illustration, consider differentiating cows and penguins in natural images; here, shape is a global semantic feature, while grass in the background is local and can appear without structure. Permuting patches via **patch randomization (PATCH-RAND)**, like in fig. 1a, corrupts global semantics while retaining local nuisances.

**N-gram randomization.** Tasks like natural language inference (NLI) — where the goal is determining if a premise sentence entails a hypothesis — satisfy the global-semantics/local-nuisances separation. Consider this example: the sentence "Bob speaks but Jon does not" contradicts "Jon speaks but Bob does not" but entails "Bob speaks". The meaning is inferred from a global structure on the words and the order they appear in. Nuisances like the number of shared words between the hypothesis and the premise predict entailment, but do not impose order (McCoy et al., 2019). Here, randomizing the order of the words corrupts the semantics: For example, one order randomization of the sentence "Jon speaks but Bob does not" is "Bob speaks but Jon does not"; the former entails "Jon speaks" but the latter contradicts it. Local nuisances, like the number of overlapping words or the presence of a certain negation word, are preserved after order randomization. We randomize the order by permuting different $n$-grams in each sentence; we call this **n-gram randomization (NGRAM-RAND)**.

### 3.2 SEMANTIC CORRUPTIONS VIA MASKING FOR A LOCATION-BASED SEPARATION

The second separation we use to build semantic corruptions is based on when a certain subset of the covariates contain a necessary part of the semantic information, masking, by removing that subset or setting it to a constant, corrupts semantics. Such masking retains nuisances outside the subset. Formally, we corrupt the semantics by replacing subsets $\mathbf{x}_S$ with a value that is out of support: for example, in images where pixels lie in $(0, 1)$, we corrupt $\mathbf{x} = [\mathbf{x}_S, \mathbf{x}_{-S}]$ as $\mathbf{x}_{\text{corrupted}} = [0 * \mathbf{x}_S, \mathbf{x}_{-S}]$.

As an illustrative example, consider a family $\mathcal{F} = \{p_D\}_{D \in R}$ with varying nuisance-label relationships. With $\mathbf{a}, \mathbf{b}$ being random uniform binary random variables, $\mathbf{e}(\rho)$ as the exponential distribution with parameter $\rho$, and $s_+(u) = \log(1 + \exp(u))$ as soft-plus, let sampling from $p_D(\mathbf{y}, \mathbf{z}, \mathbf{x})$ be:

$$\mathbf{y} = \mathbf{a} \oplus \mathbf{b}, \quad \mathbf{z} \sim \mathbf{e}(s_+(D * (2\mathbf{y} - 1))) \quad \mathbf{x} = [(2\mathbf{a} - 1)\mathbf{z}, (2\mathbf{b} - 1)\mathbf{z}].$$

For such a family, we show that masking out coordinate $\mathbf{x}_1$ is a semantic corruption: $T(\mathbf{x}) = [0, \mathbf{x}_2]$ satisfies $T(\mathbf{x}) \perp\!\!\!\perp \mathbf{y} \mid \mathbf{z}$ and $T(\mathbf{x}) \not\!\perp\!\!\!\perp \mathbf{z}$. First, due to $\mathbf{y}$ being computed as an XOR function of $\mathbf{a}, \mathbf{b}$, it holds that $\mathbf{b} \perp\!\!\!\perp \mathbf{y}$. Then, due to $\mathbf{z}$ only relying on $\mathbf{y}$ and exogenous noise, $\mathbf{b} \perp\!\!\!\perp (\mathbf{y}, \mathbf{z})$ which implies $\mathbf{b} \perp\!\!\!\perp \mathbf{y} \mid \mathbf{z}$. Given $\mathbf{z}$, $\mathbf{b}$ determines $\mathbf{x}_2$, meaning that $\mathbf{b} \perp\!\!\!\perp \mathbf{y} \mid \mathbf{z} \implies \mathbf{x}_2 \perp\!\!\!\perp \mathbf{y} \mid \mathbf{z} \implies T(\mathbf{x}) \perp\!\!\!\perp \mathbf{y} \mid \mathbf{z}$. Second, the magnitude of the second coordinate of $T(\mathbf{x})$ is $\mathbf{z}$: $|T(\mathbf{x})_2| = \mathbf{z} \implies T(\mathbf{x}) \not\!\perp\!\!\!\perp \mathbf{z}$.

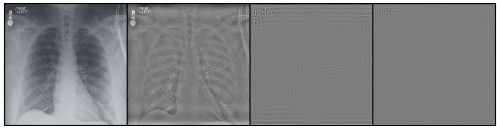 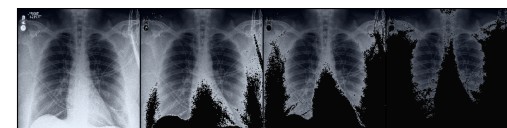

**(a)** Corruption via FREQ-FILTER. Original image to the left followed zeroing out $14, 56, 112$ of the lowest frequencies. The heart features are corrupted at $56$ but the token in the top left corner is preserved.

**(b)** Corruption via INT-FILTER. Original image to the left followed zeroing out pixels with intensities above the $80\%, 60\%, 40\%$. The heart features are corrupted at $40\%$ but non-heart parts are not.

**Figure 2:** Semantic corruptions of chest X-rays via FREQ-FILTER and INT-FILTER respectively.

**Region-of-interest-masking for object recognition.** Semantics in images can often be localized to regions-of-interest. For example, in detecting cardiomegaly, the region-of-interest is the middle of the chest where the heart resides. Masking out the region of interest removes centrally located semantic information from the chest X-ray (fig. 1b). However, nuisances like hospital-specific information (like alignment tokens (Zech et al., 2018)) are present in the border; see top right corner of the x-rays in fig. 1b. We call this **ROI-MASK**. When the region-of-interest (ROI) is centrally located, the size of the mask can be varied and the resulting border-only images are semantic corruptions.

**Premise-masking for NLI.** Semantic features in NLI rely on the meanings of the premise and the hypothesis sentences: for example, the premise states the occurrence of an event ("Alice sat while bob stood.") which can entail ("Alice sat.") or contradict ("Bob sat.") the hypothesis. The information about the setup in the premise is therefore crucial to differentiate between entailment and contradiction. If the context given by the premise is blocked out, the hypothesis sentence can predict the label only due to nuisances like the presence of negation words that correlate with contradictions (Gururangan et al., 2018). Thus, masking the premise is a semantic corruption for NLI that retains nuisance information present in the hypothesis; we call this **PREM-MASK**.

### 3.3 SEMANTIC CORRUPTIONS FOR IMAGES VIA FREQUENCY AND INTENSITY SEPARATIONS

PATCH-RAND relies on differences in relative size and ROI-MASK relies on differences in spatial position. We consider two aspects of the image that are not spatial, frequency and pixel-intensity, and give corruptions for features that depend on these aspects. Under a frequency-based separation, semantics and nuisances appear as signals of different frequencies. Under an intensity-based separation, semantics appear brighter or dimmer than nuisances. For example, consider detecting Cardiomegaly from chest X-rays, where the heart appears as an object formed of bright pixels with little variation in intensity across the pixels; the latter suggests that the heart features are low-frequency signals. However, in chest X-rays nuisances like scanner noise have low pixel-intensity and nuisances like alignment-tokens are high-frequency signals.

This observation motivates corruptions along the axes of frequency and pixel-intensity: **frequency filtering (FREQ-FILTER)** and **intensity filtering (INT-FILTER)**. FREQ-FILTER zeroes-out frequencies in the discrete fourier domain, while INT-FILTER zero-out pixels based on their intensities. See fig. 2 for how FREQ-FILTER and INT-FILTER corrupt the heart region. FREQ-FILTER and INT-FILTER are based on positional separations in frequency and intensity spaces; this is in contrast to ROI-MASK that is based on positional separations in pixel space.

### 3.4 USING SEMANTIC CORRUPTIONS IN PRACTICE

For each of the methods in table 1, we use a semantic corruption $T(\mathbf{x})$ in building a biased model $p_{tr}(\mathbf{y} \mid T(\mathbf{x}))$. For reweighting-NURD with semantic corruptions, we replace $p_{tr}(\mathbf{y} \mid \mathbf{z})$ with $p_{tr}(\mathbf{y} \mid T(\mathbf{x}))$ for a semantic corruption $T(\mathbf{x})$, for DFL and POE, we replace the model $p_{tr}(\mathbf{y} \mid \mathbf{z})$ with $p_{tr}(\mathbf{y} \mid T(\mathbf{x}))$, and for JTT, we use $p_{tr}(\mathbf{y} \mid T(\mathbf{x}))$ as the identification model.

**Choosing the "size" in PATCH-RAND, NGRAM-RAND, ROI-MASK.** For PATCH-RAND, NGRAM-RAND, and ROI-MASK, misspecifying the size parameter, such as setting it be too large or too small, runs the risk of either retaining semantics or corrupting nuisances. For example, with large patches, PATCH-RAND may not corrupt semantics while with small patches (like pixels) PATCH-RAND may corrupt all features. The resulting biased model can rely on semantics or ignore some nuisances; such biased models may not help mitigate spurious correlations.

We select the corruption parameters using a small evaluation set from the test distribution for NURD, POE, and DFL; this is common practice in OOD generalization (Mahabadi et al., 2019; Gulrajani and Lopez-Paz, 2020). For JTT, we follow Liu et al. (2021) and use nuisance annotations in the validation data. We report the results for all corruption parameters in appendix B.2.

## 4 EXPERIMENTS

In this section, we study semantic corruptions in powering NURD (Puli et al., 2022), JTT (Liu et al., 2021), and POE and DFL (Mahabadi et al., 2019). To be faithful to the original evaluations of each method, we run them on tasks from their respective papers: NURD on waterbirds, JTT on waterbirds and NLI where the nuisance is the presence of a negation word, and POE and DFL on NLI evaluated on a challenging test dataset, HANS (McCoy et al., 2019). Further, we run NURD on chest X-rays but focus on the task of Cardiomegaly detection as it is an object recognition task, instead of the original Pneumonia detection (Puli et al., 2022) which is a texture recognition task. See appendix B for implementation details. We have released the code here: ▮▮▮.

**Methods, metrics and model selection.** For images, we corrupt semantics with PATCH-RAND, ROI-MASK, FREQ-FILTER, and INT-FILTER. For text, we corrupt semantics with NGRAM-RAND and PREM-MASK. To point to the value of *semantic* corruptions relative to existing data augmentations, we also build biased models with two baseline transformations that corrupt features: random cropping (RAND-CROP) and adding gaussian noise (GAUSS-NOISE), both common data augmentations (Shorten and Khoshgoftaar, 2019). We report the average test accuracy for every method. To be able to compare to what JTT is trained for in Liu et al. (2021), we report test worst-group accuracy for JTT. For each method, we compare the performance of the original method to that of the methods run with semantic corruption (including the baselines). For every method being run with semantic corruptions, group annotations and nuisances are unavailable in the training data. Known-nuisance versions of POE, DFL, and NURD use direct knowledge of one or more nuisances during training. Vanilla JTT and JTT with corruptions use validation group annotations to early stop and tune hyperparameters. For other methods, we select the size parameter alone on a small evaluation set from the test distribution. This follows the common practice of using limited queries on a small test dataset to tune hyperparameters (Gulrajani and Lopez-Paz, 2020).

### 4.1 OBJECT RECOGNITION TASKS

To be faithful to the original evaluations of each method, we evaluate JTT on waterbirds, and NURD on both waterbirds and detecting cardiomegaly; both tasks have images of size $224 \times 224 \times 3$. For both tasks, we use PATCH-RAND (of patch sizes $7, 14, 28, 56$), ROI-MASK (of mask sizes $112, 140, 168, 196$), FREQ-FILTER (of high-pass filter sizes $196, 168, 140, 112$), and INT-FILTER (of intensity thresholds $0.1, 0.2, 0.3, 0.4$) as semantic corruptions. For the baseline RAND-CROP, we use sizes $7, 14, 28, 56$ and for GAUSS-NOISE, we use variances $0.01, 0.25, 1, 4$. Both Puli et al. (2022) and Liu et al. (2021) use the raw waterbirds data from Sagawa et al. (2019), where the task is detecting the type of bird (water or land) from images where the background is a nuisance. Unlike Liu et al. (2021), Puli et al. (2022) process the waterbirds to get a different setup from Sagawa et al. (2019). To stay true to the original evaluations of the methods, we recreate the setups as described in their respective papers.

**NURD on waterbirds.** For NURD, we recreate the waterbirds experiment from Puli et al. (2022) where the full waterbirds data from Sagawa et al. (2019) was subsampled into training, validation, and test datasets each with label balance. However, unlike Sagawa et al. (2019), the validation data comes from the same distribution as the training data. The training and validation datasets have $90\%$ waterbirds on backgrounds with water and $90\%$ landbirds on backgrounds with land. The test data has a

**Table 2:** Mean and standard error test accuracy across 10 seeds of NURD with semantic corruptions on classifying waterbirds. *Known*-nuisance NURD uses a label for the type of background as the nuisance. Consider the gap between ERM and known-nuisance NURD. NURD with semantic corruptions PATCH-RAND, ROI-MASK, FREQ-FILTER, and INT-FILTER close $99\%, 93\%, 89\%, 82\%$ of the gap respectively. Except with INT-FILTER, they outperform ERM and NURD with RAND-CROP and GAUSS-NOISE.

| Method | test acc. |
|---|---|
| *Known*-**z** NURD | $87.2 \pm 1.0\%$ |
| ROI-MASK | $87.0 \pm 1.0\%$ |
| PATCH-RAND | $86.0 \pm 1.5\%$ |
| FREQ-FILTER | $85.2 \pm 1.2\%$ |
| INT-FILTER | $84.0 \pm 1.9\%$ |
| RAND-CROP | $81.9 \pm 1.5\%$ |
| GAUSS-NOISE | $83.9 \pm 2.2\%$ |
| ERM | $69.2 \pm 2.1\%$ |

flipped relationship. Known-nuisance NURD uses an additional label denoting the background-type as the nuisance.

Table 2 gives results. Selecting the size hyperparameter based on the average accuracy over 10 seeds on an evaluation dataset (200 samples) gives size 14 for PATCH-RAND (86.0%), size 196 for ROI-MASK (87.2%), size 196 for FREQ-FILTER (85.2%), and threshold 0.2 for INT-FILTER (84.0%). Consider the gap between ERM and known-nuisance NURD. NURD with PATCH-RAND, ROI-MASK, FREQ-FILTER, and INT-FILTER close 99%, 93%, 89%, 82% of the gap respectively. NURD with these semantic corruptions outperforms ERM (69.2%) and NURD with RAND-CROP (81.9%) and GAUSS-NOISE (83.9%). In table 7 in appendix B, we give the results for all corruption parameters. These full results show that NURD with all semantic corruptions outperforms ERM (69.2%).

**JTT on waterbirds.** For JTT, we repeat the waterbirds experiment from Liu et al. (2021) which uses the original data from (Sagawa et al., 2019). The training data has 95% waterbirds on backgrounds with water and 95% landbirds on backgrounds with land. Both the validation and test datasets have bird label independent of the background. The groups here are subsets of the data that correspond to a pair of values of bird-type and background-type. Like vanilla JTT, we use group annotations in the validation data to compute worst-group error and early stop training when using PATCH-RAND and ROI-MASK. The results for vanilla JTT are from our run using the optimal hyperparameters from Liu et al. (2021).

Table 3 gives results. Selecting the corruption hyperparameters on the validation worst-group accuracy gives size 14 for PATCH-RAND (89%), size 196 for ROI-MASK (88.2%), size 112 for FREQ-FILTER (87.2%), and threshold 0.4 for INT-FILTER (87.0%). JTT with these semantic corruptions outperforms ERM (72.0%), vanilla JTT (86.5%), and JTT with the baseline corruptions RAND-CROP (59.2%) and GAUSS-NOISE (71.0%). Additionally, table 9 shows that JTT with PATCH-RAND and ROI-MASK outperforms JTT with the baseline corruptions and ERM at every patch/border-size.

**Table 3:** Test worst-group accuracies of JTT on waterbirds. JTT with semantic corruptions outperforms ERM, vanilla JTT, and JTT with baseline corruptions (RAND-CROP and GAUSS-NOISE).

| Method | test acc. |
|---|---|
| *Vanilla* JTT | 86.5% |
| ROI-MASK | 88.2% |
| PATCH-RAND | 89.0% |
| FREQ-FILTER | 87.2% |
| INT-FILTER | 87.0% |
| RAND-CROP | 59.1% |
| GAUSS-NOISE | 71.0% |
| ERM | 72.0% |

**NURD on detecting cardiomegaly** In chest X-ray classification, differences between hospitals, like the scanners used to produce the X-rays, are known to correlate thorasic conditions with non-physiological aspects in the image; for example, only some scanners render the air in the lungs in white (Zech et al., 2018). We consider the shape-based object recognition task of cardiomegaly (an irregularly sized heart) detection and, following Puli et al. (2022), construct a dataset by mixing two chest X-ray datasets: chexpert (Irvin et al., 2019) and MIMIC (Johnson et al., 2019). The training and validation datasets have 90% cardiomegaly images from MIMIC and 90% healthy images from chexpert, while the test data has a flipped relationship. Known-nuisance NURD uses the hospital identity as the nuisance.

See table 4 for results. Selecting the corruption parameters based on the mean accuracy over 10 seeds on an evaluation dataset (200 samples) gives size 7 for PATCH-RAND (77.5%), size 196 for ROI-MASK (77.2%), size 196 for FREQ-FILTER (75.6%), and threshold 0.1 for the INT-FILTER (73.9%). Consider the gap between ERM and known-nuisance NURD. NURD with PATCH-RAND, ROI-MASK, FREQ-FILTER, and INT-FILTER close 77%, 79%, 70%, 60% of the gap respectively. NURD with all these corruptions but INT-FILTER, outperforms ERM (62.0%) and NURD with RAND-CROP

**Table 4:** Mean and standard error of test accuracy across 10 seeds of NURD on chest X-rays. *Known*-nuisance NURD uses the hospital as the nuisance. Consider the gap between ERM and known-nuisance NURD. NURD with PATCH-RAND, ROI-MASK, FREQ-FILTER, and INT-FILTER close 77%, 79%, 70%, 60% of the gap respectively. Except INT-FILTER, NURD with semantic corruptions outperform ERM and NURD with baseline corruptions.

| Method | test acc. |
|---|---|
| *Known*-**z** NURD | 81.7 ± 0.3% |
| ROI-MASK | 77.2 ± 0.8% |
| PATCH-RAND | 77.5 ± 0.6% |
| FREQ-FILTER | 75.6 ± 1.3% |
| INT-FILTER | 73.9 ± 0.9% |
| RAND-CROP | 75.1 ± 0.9% |
| GAUSS-NOISE | 58.8 ± 3.7% |
| ERM | 62.0 ± 2.0% |

(75.1%) and GAUSS-NOISE (51.9%). We give the results for all corruption parameters in table 7 in appendix B; this table shows that NURD with PATCH-RAND and ROI-MASK, for all size parameters, outperform ERM (62.0%).

## 4.2 NATURAL LANGUAGE INFERENCE (NLI)

For methods POE, DFL, and JTT, we use the MNLI dataset (Williams et al., 2018) during training. The evaluations of these methods in their respective papers have different nuisances and, consequently, different test sets. In accordance, we describe the setup and the results separately. We use NGRAM-RAND (sizes $1, 2, 3, 4$) to produce nuisances for both setups. We do not run POE/DFL with PREM-MASK because PREM-MASK corrupts nuisances shared across the sentences that HANS tests.

**POE and DFL** For POE and DFL, we report test accuracies on the HANS dataset McCoy et al. (2019) as in Mahabadi et al. (2019). HANS was created to test the reliance of models on three known nuisances: 1) lexical overlap, 2) subsequence match, and 3) constituent matching subtrees in the parse trees. Known-nuisance POE and DFL use exact knowledge of these nuisances.

Table 5 gives the mean test accuracies over 10 seeds. For both DFL and POE, selecting the size hyperparameter based on the average accuracy on a small evaluation dataset (1000 samples) from the test distribution gives $n = 3$. With this size, POE with this size achieves 66.7%, improving over POE with known nuisances (66.3%). DFL with NGRAM-RAND of size 3, achieves 68.4%, closing 84% of the gap between ERM and known-nuisance DFL (69.3%). We give results for all $n$-gram sizes in table 10 in appendix B; this table shows that both POE and DFL beat ERM for all $n$-gram sizes.

**Table 5:** Average and standard deviation of accuracies (over 4 seeds) on the HANS dataset. The results for POE and DFL that use known nuisances are given under *known*. POE with NGRAM-RAND performs better than known-nuisance POE. DFL with NGRAM-RAND (NR) closes 84% of the gap between ERM and known-nuisance DFL. Both beat ERM.

| z | POE | DFL |
|---|---|---|
| *Known* | $66.3 \pm 0.6\%$ | $69.3 \pm 0.2\%$ |
| NR | $66.7 \pm 1.5\%$ | $68.4 \pm 1.5\%$ |
| ERM | − | 63.6%. |

**JTT** For JTT, we repeat the NLI experiment from Liu et al. (2021), where the presence of a negation word in the hypothesis sentence is the nuisance. The groups here are subsets of the data that correspond to a value of the label and whether or not there is a negation word in the hypothesis. Vanilla JTT uses group annotations in the validation data to tune the hyperparameters and early stop training. For each $n$-gram size, we run JTT with NGRAM-RAND for two values of the number of epochs of training for the identification model: $2, 3$. Following the hyperparameter selection procedure from Liu et al. (2021), for each $n$-gram size, we give the results for the run with the higher validation worst-group accuracy. We run *vanilla* JTT using the reported optimization hyperparameters from (Liu et al., 2021).

**Table 6:** Worst-group and average test accuracies of JTT on NLI. JTT with PREM-MASK (PM) and NGRAM-RAND (NR) outperforms vanilla JTT and ERM.

| | Worst-group | Avg. |
|---|---|---|
| *Vanilla* JTT | 71.3% | 79.1% |
| JTT + PM | 72.1% | 79.9% |
| JTT + NR | 74.3% | 79.7% |
| ERM | 67.9% | − |

Table 6 gives the results. Selecting the size hyperparameter for NGRAM-RAND using validation worst-group accuracy, like Liu et al. (2021) do for JTT, gives $n = 1$ with test worst-group accuracy of 74.3%, better than vanilla JTT's 71.3%. Additionally, table 11 shows that JTT using NGRAM-RAND at *every* size or PREM-MASK perform better than both vanilla JTT (71.3%) and ERM (67.9%).

## 5 RELATED WORK

Spurious-correlation avoiding methods (SCAMs) like (Veitch et al., 2021; Clark et al., 2019; Puli et al., 2022; He et al., 2019; Makar et al., 2022) assume the nuisance is available as additional knowledge during training. Semantic corruptions offer a complementary approach to hand-crafting nuisances or obtaining auxiliary labels, by capturing every nuisance that are separated from semantics (e.g. local nuisances and global semantics). Liu et al. (2021) build the identification model in JTT with ERM. When the identification model relies on semantics, upweighting misclassified samples produces data with a different label-semantic relationship from the one in the training data. Models trained on such data are suboptimal on test data with the same semantic relationship as the training data. Using semantic corruptions reduces the identification model's reliance on the semantics.

Sinha et al. (2021) use techniques like PATCH-RAND to restrict supports in self-supervised learning and generative modeling. Carlucci et al. (2019) use PATCH-RAND images to encourage a model to recover semantic structure. In contrast, we use PATCH-RAND to corrupt semantics and build biased models that rely on the nuisances, which help build predictive models that avoid reliance on nuisances. Work like (He et al., 2019; Puli et al., 2022) also use semantic corruptions without pointing out the reliance on knowledge about semantic features in producing nuisances. He et al. (2019) use the hypothesis as a nuisance to build a biased model for NLI; this is the masking based semantic corruption PREM-MASK. Puli et al. (2022) focus on chest X-ray classification, and use the out-of-body border of the X-ray as a nuisance; this is corrupting semantics via ROI-MASK.

Work like Bahng et al. (2020) uses CNNs with small receptive fields (RFs), to help capture local nuisances. However, their method is typically limited to very small filters because at size 3x3, deep neural networks like VGG detect non-local semantics like shapes. In contrast, the size choice in PATCH-RAND has no bearing on the choice of the model; we used default vision models. Bras et al. (2020) automatically identify and remove examples with nuisances using adversarial filtering, but risk removing genuinely easy examples. Qin et al. (2021) work solely with vision transformers and consider why labels can be predicted from transformations akin to patch-randomized images. Concluding that this can only be due to nuisances, they propose to encourage the transformer to have predictions and representations of the original images be dissimilar from those of patch-randomized images. In contrast, our work applies to general flexible models and shows that semantic corruptions can be used to break the label's relationship with nuisances in the original images.

## 6 DISCUSSION

We study the use of semantic knowledge in building robust models. Given a procedure to corrupt semantics, anything that predicts the label in the corrupted input is a nuisance. Using semantic corruptions, practitioners can run different kinds of spurious-correlation avoiding methods (SCAMs) (NURD, JTT, DFL, POE). With these semantic corruptions, methods like NURD and DFL perform close to how they would with known nuisances, and methods like JTT perform better than how they would when relying on ERM on the raw covariates to build a nuisance.

**Limitations.** The quality of any semantic corruption, and thus the quality of the results, depends on the extent to which semantics are destroyed and nuisances are retained. PATCH-RAND and NGRAM-RAND are built to corrupt global semantics and retain local nuisances, ROI-MASK to retain nuisances outside the ROI and PREM-MASK to retain nuisances in hypothesis.

When applied to cases other than what they were built for, these methods may not destroy all the semantics or retain all the nuisances and thus yield models that generalize. For example, when PATCH-RAND is used blindly on covariates with local semantics, the biased model may rely on said semantics; this in turn guides the predictive model to ignore these semantics and, thus, lose predictive performance. Alternatively, when nuisances are global, PATCH-RAND may corrupt them. For example in detecting cows and penguins, other nuisance animals (like dogs) may co-occur with cows more often; PATCH-RAND would corrupt this nuisance animal. Using PATCH-RAND in a SCAM for such tasks could lead to non-robust predictive models that rely on corrupted nuisances.

Our experiments show blind usage does not always lead to poor performance despite violations of the separation that underlies the semantic corruption. In both classifying waterbirds and NLI, there exist local semantics, like small beaks and individual words, that are not corrupted by PATCH-RAND and NGRAM-RAND respectively. However, in our Waterbirds and NLI experiments, we show models built using semantic corruptions close more than $80\%$ of the gap in test performance between ERM and the methods that use known nuisances. Similarly, ROI-MASK corrupts nuisances in the ROI and retains semantics outside it. However, in both waterbirds and cardiomegaly detection, where nuisances like hospital-specific tokens and background features lie in the ROI, ROI-MASK still helps NURD close $> 75\%$ of the gap in test performance between ERM and NURD with a known nuisance.

**Summary.** Semantic corruptions power SCAMs to build models robust to spurious correlations without requiring extra annotations in the form of known nuisances during training or relying on hard to verify assumptions like ERM-trained models relying on nuisances. As discussed above, our experiments point out that using semantic corruptions leads to improved robustness even under violations of the separation assumptions they are built off of. These two properties indicate the value of semantic corruptions as a way to build robust models.

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

## A    FURTHER DETAILS ABOUT SPURIOUS-CORRELATION AVOIDING METHODS

**NURD.**    Focusing on mitigating spurious correlations, Puli et al. (2022) identify a conditional that has performance guarantees on every test distribution within a family of distributions with varying nuisance-label relationships: $p_{te} \in \mathcal{F}$. They develop NURD to learn the conditional using data only from $p_{tr} \neq p_{te}$. NURD uses 1) the *nuisance-randomized distribution*, $p_{\perp}(\mathbf{y}, \mathbf{z}, \mathbf{x}) = p(\mathbf{y})p_{\perp}(\mathbf{z})p(\mathbf{x} \mid \mathbf{y}, \mathbf{z})$, where $\mathbf{z} \perp\!\!\!\perp_{p_{\perp}} \mathbf{y}$, and 2) an *uncorrelating representation* $r(\mathbf{x})$ for which $\mathbf{z} \perp\!\!\!\perp_{p_{\perp}} \mathbf{y} \mid r(\mathbf{x})$. NURD builds models of the form $p_{\perp}(\mathbf{y} \mid r(\mathbf{x}))$ using $r(\mathbf{x})$ that are most informative of the label.

We run reweighting-NURD, which uses a biased model $p_{tr}(\mathbf{y} \mid \mathbf{z})$ as an importance weight to compute loss under the nuisance-randomized distribution: $p_{\perp}(\mathbf{y}, \mathbf{z}, \mathbf{x}) = \frac{p_{tr}(\mathbf{y})}{p_{tr}(\mathbf{y} \mid \mathbf{z})} p_{tr}(\mathbf{y}, \mathbf{z}, \mathbf{x})$.

To run reweighting-NURD with semantic corruptions, we replace $p_{tr}(\mathbf{y} \mid \mathbf{z})$ with $p_{tr}(\mathbf{y} \mid T(\mathbf{x}))$ for a semantic corruption $T(\mathbf{x})$. Semantic corruptions are noisy functions of $\mathbf{x}$: with noise $\epsilon$ such that $(\mathbf{y}, \mathbf{z}, \mathbf{x}) \perp\!\!\!\perp_{p_D} \epsilon$, $T(\mathbf{x}) = U(\mathbf{x}, \epsilon)$. This implies

$$\mathbf{y} \perp\!\!\!\perp_{p_{\perp}} \epsilon \mid \mathbf{x} \implies \mathbf{y} \perp\!\!\!\perp_{p_{\perp}} \mathbf{x}, \epsilon \mid \mathbf{x} \implies \mathbf{y} \perp\!\!\!\perp_{p_{\perp}} T(\mathbf{x}) \mid \mathbf{x}$$

Thus, $r(\mathbf{x}) = \mathbf{x}$ is uncorrelating and $p_{\perp}(\mathbf{y} \mid \mathbf{x})$ achieves the optimality guarantees in Puli et al. (2022). These optimality guarantees imply that regardless of the test nuisance-label relationship, $p_{\perp}(\mathbf{y} \mid \mathbf{x})$ will achieve optimal performance within the class of models like $p_{\perp}(\mathbf{y} \mid r(\mathbf{x}))$.

**End-to-end bias mitigation.**    Mahabadi et al. (2019) consider two methods to train a *biased* model $p_{tr}(\mathbf{y} \mid \mathbf{z})$ and a base predictive model jointly to make the base model predict without relying on the biases. The methods use and fine-tune a BERT model (Devlin et al., 2019) and do not propagate the gradients of the biased model to update the common parameters (token embeddings in this case). They propose 1) POE, where the `log` of the product of the predictions (the output probabilities) of the two models is used to compute the classification loss and 2) DFL, where the biased model is used to weight the cross-entropy loss for the base model.

The intuition for POE is that the samples for which the biased model classifies correctly will not contribute to the gradients of the base model; thus the base model focuses more on classifying samples that the biased model misclassifies. The DFL algorithm weights each sample as the biased model's predicted probability of all but the label, exponentiated with $\gamma > 0$. This downweights samples that the biased model classifies correctly which in turn mitigates the base model's reliance on a nuisance which only helps predict the downweighted samples correctly.

Mahabadi et al. (2019) build the biased model as $p_{tr}(\mathbf{y} \mid \mathbf{z})$ with known nuisances $\mathbf{z}$. We replace this model with $p_{tr}(\mathbf{y} \mid T(\mathbf{x}))$ for a semantic corruption $T(\mathbf{x})$.

**Just Train Twice (JTT).**    JTT works in two stages: 1) build an "identification" model via ERM on the training data to isolate samples that are misclassified due to reliance on the nuisances and 2) train a model via ERM on data with the loss for the misclassified samples upweighted (by constant $\lambda$).

The identification model in JTT is built to be a biased model. When the identification model equals $p_{tr}(\mathbf{y} \mid \mathbf{z})$, it exactly misclassifies the samples in the groups in the minority group[1]. Upweighting these samples produces a dataset with lesser dependence between the nuisance and the label. Then, models learned on the upweighted data depend more on the semantic features.

In this work, we build the identification model on semantic corruptions i.e. we learn $p_{tr}(\mathbf{y} \mid T(\mathbf{x}))$. The training samples to be upweighted are the ones misclassified when predicting with the identification model on semantic-corrupted versions of the sample, i.e. $T(\mathbf{x})$. The second stage is run as in (Liu et al., 2021) with training data.

## B    FURTHER EXPERIMENTAL DETAILS

See fig. 3 for an example of PATCH-RAND for chest X-rays.

---

[1]The minority group is the set of samples that the nuisance misclassifies. For example, when $p_{tr}(\mathbf{y} = \mathbf{z}) > p_{tr}(\mathbf{y} \neq \mathbf{z})$, then the minority group is the set of samples with $\mathbf{y} \neq \mathbf{z}$ because using only the nuisance feature results in predicting $\mathbf{y} = b$ on samples with $\mathbf{z} = b$.

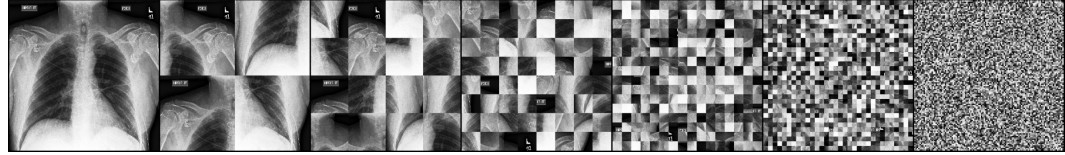

**Figure 3:** Example of PATCH-RAND of a chest X-ray image. The original image is the left most followed by PATCH-RANDs with sizes $112, 56, 28, 14, 7, 2$ respectively.

## B.1 IMPLEMENTATION DETAILS

Each experiment in the paper was run on up to 2 RTX8000 GPUs. The hyperparameters for methods which use known nuisances in the training data like NURD, POE, DFL are tuned on validation data from the training distribution. We do the same when using semantic corruptions.

**Experimental details for Waterbirds**  For the NURD setup, the training, validation, and test datasets have $3020, 756, 800$ samples respectively. We use a single architecture to parameterize the predictive model and the weight model in this experiment: two fully connected layers on top of a ResNet18 initialized at weights pretrained on Imagenet. We use the same training procedure for NURD with known nuisances or with semantic corruptions. Both models are trained with cross-entropy. The weight model is optimized with the default Adam optimizer for 20 epochs with a batch size of 64. The predictive model is optimized with the Adam optimizer for 20 epochs with a learning rate of $0.0002$, a weight decay of $0.01$, and a batch size of 250.

For the JTT setup, the training, validation, and test datasets have $4795, 1199, 5794$ samples respectively. For JTT, we use the same model and model parameters as Liu et al. (2021) using their released code. We repeat the details here for completeness. The model for both stages of JTT is a ResNet-50. Both models are optimized by stochastic gradient descent (SGD) with momentum $0.9$, weight decay $1.0$, and learning rate $1 \times 10^{-5}$. Both models are trained for 300 epochs with batch size 64, using batch normalization and no data augmentation. The identification model used to select samples to upweight corresponds to epoch 60 and the upweighting constant is $\lambda = 100$.

**Experimental details for cardiomegaly detection.**  The training, validation, and test datasets are fixed across seeds and have $18000, 2000, 1000$ samples respectively. To run reweighting-NURD, we use a single architecture to parameterize the predictive model and the weight model in this experiment: two fully connected layers on top of a ResNet18 initialized at weights pretrained on Imagenet. In known-nuisance NURD with the hospital as the nuisance, the biased model is an estimate of $p_{tr}(\mathbf{y} \mid \text{hospital})$, which is obtained by binning the samples based on the hospital and averaging the labels. We use the same training procedure for NURD with known nuisances or with semantic corruptions. Both weight and predictive models are trained with cross-entropy. The weight model and the predictive model are optimized with the Adam optimizer over 25 epochs with a batch size of 256, and learning rate $0.001$.

**Implementation details for NLI**  For POE and DFL, we build classifiers by fine-tuning a pretrained BERT model (Devlin et al., 2019) on the data. We follow the same training procedure and hyper-parameter details as used in Mahabadi et al. (2019) — models were trained on the MNLI training dataset which consists of 392k examples, with a learning rate of $2 \times 10^{-5}$ with a batch size of 8 using the Adam Optimizer. All models are trained for 3 epochs. The development set contains 9815 examples and the HANS test contains 30000 examples. Since the HANS dataset has only two labels — 'entailment' and 'non-entailment' — we combine the neutral and contradiction classes during inference on HANS.

For the JTT setup, Liu et al. (2021) mix the training and development sets from MNLI and create their own training, validation, and test sets of sizes $206175, 82462, 123712$ respectively. For JTT, we use the same model and model parameters as Liu et al. (2021) using their released code. We use the optimal hyperparameters reported in Liu et al. (2021) for the learning rate, weight decay, and the upweighting constant. We repeat the details here for completeness. The model for both stages of JTT is a pretrained BERT model that is finetuned during training. Both models are optimized by the AdamW optimizer with clipping for the predictive model, no weight decay, and an initial learning rate of $2 \times 10^{-5}$. Both models are trained for 5 epochs with batch size 32 and dropout. The identification model used to select samples to upweight corresponds to epoch 2 for vanilla JTT (reported optimal in Liu et al. (2021)); for JTT with semantic corruption, we select one from

**Table 7:** Mean and standard error of test accuracy across 10 seeds of NURD on classifying water-birds. *Known*-nuisance NURD uses a label for the type of background as the nuisance. Selecting the size hyperparameter based on the average accuracy over 10 seeds on an evaluation dataset (200 samples) gives 14 for PATCH-RAND, 196 for ROI-MASK, 196 for FREQ-FILTER, and 0.2 for INT-FILTER. Consider the gap between ERM and known-nuisance NURD. NURD with PATCH-RAND, ROI-MASK, FREQ-FILTER, and INT-FILTER close $99\%, 93\%, 89\%, 82\%$ of the gap respectively. NURD with these semantic corruptions outperforms ERM and NURD with RAND-CROP and GAUSS-NOISE. NURD with all semantic corruptions outperforms ERM ($69.2\%$).

|  | *known* z | RM 196 | RM 168 | RM 140 | RM 112 | PR 7 | PR 14 | PR 28 | PR 56 | ERM |
|---|---|---|---|---|---|---|---|---|---|---|
| Mean | 87.2% | 87.0% | 87.6% | 85.1% | 85.5% | 82.6% | 86.0% | 83.9% | 80.4% | 69.2% |
| Std. err. | 1.0% | 1.0% | 1.8% | 1.0% | 1.6% | 2.2% | 1.5% | 1.2% | 2.0% | 2.1% |

|  |  | FF 196 | FF 168 | FF 140 | FF 112 | IF 0.1 | IF 0.2 | IF 0.3 | IF 0.4 |  |
|---|---|---|---|---|---|---|---|---|---|---|
| Mean |  | 85.2% | 82.6% | 83.7% | 80.1% | 83.1% | 84.0% | 80.7% | 78.7% |  |
| Std. err. |  | 1.2% | 1.6% | 1.0% | 1.5% | 1.8% | 1.9% | 3.3% | 3.4% |  |

|  |  | CROP 56 | CROP 28 | CROP 14 | CROP 7 | GAUSS 0.01 | GAUSS 0.25 | GAUSS 1 | GAUSS 4 |  |
|---|---|---|---|---|---|---|---|---|---|---|
| Mean |  | 81.9% | 80.3% | 74.9% | 67.9% | 75.8% | 74.1% | 78.0% | 83.9% |  |
| Std. err. |  | 1.5% | 1.3% | 1.5% | 2.9% | 3.2% | 3.1% | 3.4% | 1.4% |  |

$2, 3$ using validation group annotations. For both, the upweighting constant is $\lambda = 6$. Our runs with these parameters did not yield the test worst-group accuracy reported in (Liu et al., 2021) ($72.6\%$); our experiments yielded a test worst-group accuracy $71.3\%$. We expect this may be due to the differences in the random seed; JTT is sensitive to hyperparameters and differences in order of batches may result in drops in performance.

In NGRAM-RAND, when the number of words in the sentence is not a multiple of $n$, there will be one $k$-gram ($k < n$). In implementing NGRAM-RAND, we ensure that the position of this k-gram is randomized i.e. we make sure that it does not always occur at the end of the sentence, for example. NGRAM-RAND is implemented before word-piece tokenization (which BERT uses), to ensure that we randomize words instead of subwords.

We also create a small HANS-like development set, which can be optionally used to tune the size parameter. This set is constructed by randomly sampling 1000 examples from the HANS training set, which does not have any overlap with the main HANS test set.

### B.2 FULL RESULTS TABLES

We give the results for all size parameters; see table 7, table 8, table 9, table 11, and table 10. To report the same metrics as in Mahabadi et al. (2019) for POE and DFL and Puli et al. (2022) for NURD, we report standard error for NURD and standard deviation for POE and DFL .

**Table 10:** Average accuracies and standard deviation over 4 seeds of POE and DFL with semantic corruptions on the HANS dataset. The results for *known* POE and DFL from Mahabadi et al. (2019), where both methods use known nuisances. For both methods, selecting the size hyperparameter based on the average accuracy on a small evaluation dataset (1000 samples) from the test distribution gives $n = 3$. With this size, POE with NGRAM-RAND performs better than known-nuisance POE while DFL with NGRAM-RAND closes $84\%$ of the gap between ERM and known-nuisance DFL .

| z | POE | DFL |
|---|---|---|
| *Known* | $66.3 \pm 0.6\%$ | $69.3 \pm 0.2\%$ |
| 1-gram | $65.7 \pm 2.0\%$ | $66.5 \pm 1.5\%$ |
| 2-gram | $66.0 \pm 0.9\%$ | $68.5 \pm 0.7\%$ |
| 3-gram | $66.7 \pm 1.5\%$ | $68.4 \pm 1.5\%$ |
| 4-gram | $66.2 \pm 2.9\%$ | $65.0 \pm 2.0\%$ |
| ERM | − | $63.6\%$. |

**Table 8:** Mean and standard error of test accuracy across 10 seeds of NURD on detecting cardiomegaly from chest X-rays. *Known*-nuisance NURD uses the hospital as the nuisance. Selecting the corruption parameters based on the mean accuracy over 10 seeds on an evaluation dataset (200 samples) gives 7 for PATCH-RAND, 196 for ROI-MASK, 196 for FREQ-FILTER, and 0.1 for the INT-FILTER. Consider the gap between ERM and known-nuisance NURD. NURD with PATCH-RAND, ROI-MASK, FREQ-FILTER, and INT-FILTER close $77\%, 79\%, 70\%, 60\%$ of the gap respectively. NURD with PATCH-RAND, ROI-MASK, FREQ-FILTER outperforms ERM and NURD with RAND-CROP and GAUSS-NOISE. NURD with PATCH-RAND and ROI-MASK outperforms ERM for all size parameters.

|  | *known* z | RM 196 | RM 168 | RM 140 | RM 112 | PR 7 | PR 14 | PR 28 | PR 56 | ERM |
|---|---|---|---|---|---|---|---|---|---|---|
| Mean | 81.7% | 77.2% | 75.0% | 74.4% | 68.9% | 77.5% | 75.2% | 72.2% | 69.2% | 62% |
| Std. err. | 0.3% | 0.8% | 0.9% | 1.3% | 1.6% | 0.6% | 0.8% | 1.3% | 2.3% | 2.0% |

|  | FF 196 | FF 168 | FF 140 | FF 112 | IF 0.1 | IF 0.2 | IF 0.3 | IF 0.4 |
|---|---|---|---|---|---|---|---|---|
| Mean | 75.6% | 74.2% | 70.4% | 71.4% | 73.9% | 69.8% | 61.8% | 54.0% |
| Std. err. | 1.3% | 1.5% | 2.0% | 1.4% | 0.9% | 1.5% | 3.2% | 2.3% |

|  | CROP 56 | CROP 28 | CROP 14 | CROP 7 | GAUSS 0.01 | GAUSS 0.25 | GAUSS 1 | GAUSS 4 |
|---|---|---|---|---|---|---|---|---|
| Mean | 75.1% | 71.2% | 66.1% | 67.9% | 58.8% | 49.0% | 51.9% | 51.1% |
| Std. err. | 0.9% | 1.2% | 1.7% | 1.9% | 3.7% | 2.8% | 2.1% | 2.3% |

**Table 9:** Test worst-group accuracies of JTT with semantic corruptions on waterbirds. Selecting the corruption hyperparameters on the validation worst-group accuracy gives size 14 for PATCH-RAND, size 196 for ROI-MASK, size 112 for FREQ-FILTER, and threshold 0.4 for INT-FILTER. JTT with these semantic corruptions outperforms ERM, vanilla JTT, and JTT with the baseline corruptions RAND-CROP and GAUSS-NOISE. JTT with PATCH-RAND and ROI-MASK outperforms JTT with the baseline corruptions and ERM at every patch/border-size.

| *Vanilla* JTT | RM 196 | RM 168 | RM 140 | RM 112 | PR 7 | PR 14 | PR 28 | PR 56 | ERM |
|---|---|---|---|---|---|---|---|---|---|
| 86.5% | 88.2% | 88.0% | 86.9% | 86.2% | 89.3% | 89.0% | 88.9% | 89.1% | 72% |

| FF 196 | FF 168 | FF 140 | FF 112 | IF 0.1 | IF 0.2 | IF 0.3 | IF 0.4 |
|---|---|---|---|---|---|---|---|
| 82.5% | 84.5% | 85.2% | 87.2% | 69.1% | 80.0% | 81.7% | 87.0% |

| CROP 56 | CROP 28 | CROP 14 | CROP 7 | GAUSS 0.01 | GAUSS 0.25 | GAUSS 1 | GAUSS 4 |
|---|---|---|---|---|---|---|---|
| 59.1% | 0.0% | 0.0% | 0.0% | 0.0% | 0.0% | 71.0% | 0.0% |

**Table 11:** Worst-group and average test accuracies of JTT with semantic corruptions on NLI. JTT with PREM-MASK and NGRAM-RAND of every size outperforms vanilla JTT. Selecting the size hyperparameter for NGRAM-RAND using validation worst-group accuracy, like Liu et al. (2021) do for vanilla JTT, gives $n = 1$. At this size, JTT with NGRAM-RAND outperforms vanilla JTT by $3\%$.

|  | Worst-group | Average |
|---|---|---|
| *Vanilla* JTT | 71.3% | 79.1% |
| PREM-MASK | 72.1% | 79.9% |
| 1-gram | 74.3% | 79.7% |
| 2-gram | 71.9% | 80.0% |
| 3-gram | 72.0% | 80.1% |
| 4-gram | 73.4% | 80.4% |
| ERM | 67.9% | – |

