# OpenReview forum: "Nuisances via Negativa: Adjusting for Spurious Correlations via Data Augmentation"
_ICLR.cc/2023/Conference — Submitted to ICLR 2023_

### Official Review · Reviewer_yHka · 2022-10-24

**Confidence:** 4
**Clarity, Quality, Novelty And Reproducibility:** The presentation of this paper is cle…
**Correctness:** 3
**Technical Novelty And Significance:** 3
**Empirical Novelty And Significance:** 3
**Recommendation:** 6

**Strength And Weaknesses:**

Strengths:

S1: This paper focuses on adjusting for spurious correlations, which is an important problem in reliable machine learning.

S2: The main idea of adopting data augmentations corrupt semantic information to produce models that identify and adjust for spurious correlations is reasonable.

S3: The experimental results and analysis look good.

Weaknesses:

W1: A few details are not very clear. How to control the data augmentation directions? How to identify and disentangle the corrupt semantic features.

W2: I wonder what the semantic augmented data look like and it may be better to visualize them.

W3: It might be interesting if the authors can provide experimental results on large-scale datasets like ImageNet with spurious cues.

**Summary Of The Paper:**

This work proposes a new method to produce robust models by data augmentations that corrupt semantic information. Semantic corruptions power nuisance-avoiding methods to build robust models against spurious correlations without requiring extra annotations or strong assumptions. This paper analyzes semantic corruptions in powering different robust-modeling methods for multiple out-of-distribution tasks.

**Summary Of The Review:**

This work is focusing on an important problem in reliable machine learning. The motivation of this paper is reasonable and the experimental results look good but there are a few detailed issues that need to be clarified.

---

### Official Review · Reviewer_fx5t · 2022-10-24

**Confidence:** 4
**Correctness:** 3
**Technical Novelty And Significance:** 3
**Empirical Novelty And Significance:** 3
**Recommendation:** 6

**Clarity, Quality, Novelty And Reproducibility:**

The paper is quite well written and clear. It is also technically sound.
The contributions is fairly novel but limited to methological aspects.
I might have missed it, but I haven't seen a link to the authors code.

**Strength And Weaknesses:**

Pros :
- The method is clear and addresses an important issue in deep learning
- The approach is tested on various tasks
- The method can be readily plugged into recent approaches from the literature

Cons :
- The method applies only to data where positional information has an impact on semantics but not on nuisance
- The methodological contribution needs to be validated by comparing to other data augmentation mechanisms that randomize inputs

**Summary Of The Paper:**

This paper introduces a data augmentation framework that attempts to avoid spurious correlations (nuisance) in order to enjoy increased generalization.
To that end, the authors propose to corrupt the semantic signal in the input. If a model maintains a good level of prediction under these circumstances, it means the model relies on nuisance signals and is thus not robust. Achieving this through data augmentation is possible if nuisance and semantic signals in the inputs are well separable. The authors argue that one way to exploit signal separation is when:
- local modifications of inputs does not change nuisance but harms semantics.
- semantic signal is position dependent (like organ positions in medical image inputs).
The authors leverage two random patch permutations (as data augmentation mechanisms) to exploit one of the above separation in practice.


**Summary Of The Review:**

Major remarks:

The paper explains how to train a biased model using a method from the literature plus one of the introduced mechanisms. But the paper is too quick on how to exploit this biased model to avoid spurious correlations. Please add more information in this regard. Appendix A does not say more than the main text.

The proposed mechanisms belong to the family of random transformation type of data augmentation. Yet in the experimental section, no comparison with such alternative data augmentation mechanisms can be found. The authors should demonstrate that the positional information on which their augmentations rely is really a key aspect or if random cropping, random erasing or others might compete.

In Table 2, RM + NURD outperforms known nuisance + NURD for RM size 168. Is this normal ?


Minor :

Std deviations wrt the reported accuracies can only be found in the appendices. If paper length permits, I think it would be preferable to display them immediately.

Page 2 : the authors start using mathematical notations that were not introduced before.

---

### Official Review · Reviewer_NEkh · 2022-10-25

**Confidence:** 2
**Clarity, Quality, Novelty And Reproducibility:** The clarity is poor. The novelty is n…
**Correctness:** 3
**Technical Novelty And Significance:** 3
**Empirical Novelty And Significance:** 3
**Recommendation:** 6

**Strength And Weaknesses:**

Strengths:
1. The proposed method is well analyzed with rich ablation studies.
2. This paper validates the effectiveness of the method on multiple tasks.


Weakness:
1.  The writing of this paper needs polishing. I have a hard time understanding the expression of this paper.
2.  to produces -> to produce. In addition, there are many such errors.
3.  It is recommended to validate the effectiveness of the method on larger datasets.

**Summary Of The Paper:**

This article mitigates OOD issues in ERM with semantic corruption.

**Summary Of The Review:**

Due to the poor writing of this article, it is difficult for me to follow what the author wants to convey.

---

### Official Review · Reviewer_QvJV · 2022-10-26

**Confidence:** 3
**Correctness:** 4
**Technical Novelty And Significance:** 3
**Empirical Novelty And Significance:** 2
**Recommendation:** 6

**Clarity, Quality, Novelty And Reproducibility:**

See section above.

**CLARITY**

The paper is overall quite clear, however:

1. Introduction and section 3.1: When describing global semantics vs local nuisances, your definition of "global" vs "local" is not very clear to me, I only understood what you meant through the examples you provided.

2. Section 2: none of the symbols used in the NURD paragraph is defined ($p_{tr}$, $p_{te}$, independence symbol, ..)


**Strength And Weaknesses:**

**STRENGTHS**

1. The paper discusses a simple yet effective to generalize and improve existing methods.

2. The performances reached in the experiments are close to then ones achieved when having labelled data for the nuisances.

3. The method seems to be robust to the choice for the "size" hyperparameter, which makes it easier to be used in practice.





**WEAKNESSES**

1. I am not very convinced on the level of novelty of the paper. While I do acknowledge that the authors present a more unified view of existing methods by formally introducing semantics/separations, in practice many of the presented ideas have been already applied to the same or similar task (for example ROI-MASK in Puli et al, 2022).

2. The two separations introduced in the paper work well with the benchmark datasets for this task. However I do not think they are generic enough in many real tasks, where there is no clear separation between semantics and nuisances, or no obvious data transformation to apply.
ROI masking is for example hard to apply in practice when relevant objects might be in very different parts of an image.



**Summary Of The Paper:**

This paper shows a simple way to improve the performances of existing methods to avoid spurious correlations, which can be applied to all methods that rely on a bias model that predicts the labels using the nuisances in the data.



The authors propose to build the biased model by using transformations to the input data that are such that nuisances are preserved, but the semantic information (which is the one that should drive the classifier) is destroyed. Two types of transformation are considered: 1) permutation-based, which can be applied when only nuisances (and not semantic information) are still present when the input data is randomized; 2) mask-based, which can be applied if only nuisances are present in a subsed of the input data.



Experiment show that the method successfully avoids spurious correlations when the biased model is constructed as explained above.

**Summary Of The Review:**

This is overall an interesting work. However I have some concerns on the novelty of the presented ideas, as well as the applicability of the method in many real world applications.

---

### Decision · Program_Chairs · 2023-01-20

**Decision:**

Reject

**Justification For Why Not Higher Score:**

technical novelty is very limited and the settings are not realistic

**Justification For Why Not Lower Score:**

n/a

**Metareview: Summary, Strengths And Weaknesses:**

This paper proposes augmentation methods that strengthen the biased model of existing spurious-correlation avoiding methods (SCAMs) to be more biased. By using the representative characteristics of semantic and nuisance (global vs local, frequency, pixel intensity), the semantic of data is intentionally reduced to induce focus on nuisance.

Four reviewers reviewed this paper, all agreed that the proposed method is simple and effective, and they suggested borderline accept. However, upon further discussion, most reviewers agreed that the method and novelty are very limited, and none of the reviewers strongly argued for acceptance of this paper. In addition, this AC has the following concerns about this submission, and it seems that additional round of review is needed because it requires considerable revision to resolve these concerns.

- The authors propose a method of learning a based model by augmentation using knowledge about nuisance without a nuisance label, but more practical issue that has been more focused in recent studies is the situation in which there is no knowledge about bias. Under this setting, the problem becomes quite simple, and hence the technical novelty is very limited.

- If the bias does not have pre-assumed global characteristics, this method cannot be applied, and experiments on various semantic/nuisance datasets are lacking.

- In order to properly remove semantics, the authors assume an evaluation set with the same distribution as the test distribution because the method is probably very sensitive to the hyperparameters. There have been such studies in the past, but it is not practical to assume it.

- it would be nice to compare it with methods such as Generalized cross entropy used in 'Learning from Failure' paper (LfF does not assume knowledge of bias)


**Summary Of Ac-Reviewer Meeting:**

Most reviewers agreed that the paper's novelty is limited. And although they did not adjust the score, all of them were of the opinion that they did not oppose the rejection of this paper. As outlined in the meta review, the novelty is limited and the setting being considered is somewhat unrealistic, so I suggest rejecting it.